# Role of the MDR Efflux Pump AcrAB in Epithelial Cell Invasion by *Shigella flexneri*

**DOI:** 10.3390/biom13050823

**Published:** 2023-05-11

**Authors:** Marco Coluccia, Aude Béranger, Rita Trirocco, Giulia Fanelli, Francesco Zanzi, Bianca Colonna, Milena Grossi, Gianni Prosseda, Martina Pasqua

**Affiliations:** Istituto Pasteur Italy, Department of Biology and Biotechnologies “Charles Darwin”, Sapienza University of Rome, 00185 Rome, Italy; marco.coluccia@uniroma1.it (M.C.); aberanger@imm.cnrs.fr (A.B.); rita.trirocco@uniroma1.it (R.T.); giulia.fanelli@uniroma1.it (G.F.); zanzi.1582366@studenti.uniroma1.it (F.Z.); bianca.colonna@uniroma1.it (B.C.); milena.grossi@uniroma1.it (M.G.); gianni.prosseda@uniroma1.it (G.P.)

**Keywords:** *Shigella*, AcrAB, efflux pumps, microbe–host cell interaction

## Abstract

The tripartite complex AcrAB-TolC is the major RND pump in *Escherichia coli* and other Enterobacteriaceae, including *Shigella*, the etiological agent of bacillary dysentery. In addition to conferring resistance to many classes of antibiotics, AcrAB plays a role in the pathogenesis and virulence of several bacterial pathogens. Here, we report data demonstrating that AcrAB specifically contributes to *Shigella flexneri* invasion of epithelial cells. We found that deletion of both *acrA* and *acrB* genes causes reduced survival of *S. flexneri* M90T strain within Caco-2 epithelial cells and prevents cell-to-cell spread of the bacteria. Infections with single deletion mutant strains indicate that both AcrA and AcrB favor the viability of the intracellular bacteria. Finally, we were able to further confirm the requirement of the AcrB transporter activity for intraepithelial survival by using a specific EP inhibitor. Overall, the data from the present study expand the role of the AcrAB pump to an important human pathogen, such as *Shigella*, and add insights into the mechanism governing the *Shigella* infection process.

## 1. Introduction

Multidrug-resistance (MDR) efflux pumps (EPs) are transmembrane transporters capable of extruding a wide range of toxic compounds, including antibiotics [1,2]. Genes encoding MDR EPs are generally located on the chromosome and are highly conserved across species. Currently, genes encoding MDR EPs are considered not only the simple result of recent evolution favored by the intense use of antibiotics but ancient genes encoding protein complexes deeply involved in the physiology of the bacterial cell [3,4]. This is further underlined by the fact that, in addition to antibiotics, MDR EPs are able to expel a large range of different molecules from the cell, including bacterial metabolites, siderophores, heavy metals, quorum-sensing molecules and virulence factors [2,5,6]. This capability makes MDR EPs important players in the maintenance of cellular homeostasis, in the interplay between bacteria, in bacteria–host cell interactions, in biofilm formation and in the virulence [7,8,9,10,11].

Bacterial MDR EPs are commonly present as single-component transporters embedded in the inner membrane. They can also form tripartite complexes that cross the double membranes in Gram-negative bacteria [12]. MDR EPs mainly belong to the following families: the ATP binding cassette (ABC) superfamily, the major facilitator superfamily (MFS), the multidrug and toxic compound extrusion (MATE) family, the small multidrug resistance (SMR) family and the resistance nodulation division (RND) family [2].

Members of the RND family play a critical role in the emergence of multidrug resistance in many Gram-negative bacterial pathogens [1,13,14]. Six MDR EPs of the RND family are usually present in *E. coli* K12, namely AcrAB, AcrD, AcrEF, MdtABC, MdtEF, and CusCFBA. The *E. coli* species is very wide and includes, besides commensal *E. coli,* several groups of pathogenic *E. coli* that can cause a broad range of diseases. Through phylogenetic analyses and genome sequencing, *Shigella*, the etiological agent of bacillary dysentery, has been shown to belong to the diverse *E. coli* species rather than forming a distinct genus [15,16]. Although it is now considered a specific group of diarrheal *E. coli* (DEC), the original name, *Shigella,* has been retained for diagnostic and epidemiological purposes [17,18,19]. Despite the high genome homology of *Shigella* and the commensal *E. coli* [16,19], most of efflux pumps of the RND family have been lost by *S. flexneri*. Indeed, we have previously shown that in *S. flexneri* M90T, four RND MDR EPs have been silenced by gene disruption (*cusB, mdtABC*, and *mdtEF*) or by large deletion (*acrEF*), leaving functional only the AcrAB and AcrD [11]. Silencing of MDR EP genes in *S. flexneri* is not surprising as many housekeeping genes present in the chromosome of the commensal *E. coli* ancestor have been lost following an intense gene decay [20], accompanied in some cases by a patho-adaptation process [21,22,23].

As for the two residual RND MDR EPs of *Shigella*, the tripartite AcrAB is considered the most relevant MDR EP of *E. coli* for its wide substrate profile and its high abundance within the cell compared to other EPs [1,24,25]. Similar to many other RND transporters, AcrB is a homotrimer. It associates with six periplasmatic AcrA molecules. Both proteins are encoded by the same *acrAB* operon. The AcrA protein plays an important role in the assembly and in the stable maintenance of the entire pump and transmits conformational changes of AcrB to TolC, facilitating the opening of the outer membrane channel [12,25]. The AcrAB-TolC EP exhibits a broad substrate profile which includes multiple classes of antibiotics, biocides, detergents, dyes and metals [1,5,25]. Moreover, AcrAB has been shown to contribute to the virulence of several human and plant pathogens [8,26,27,28].

The AcrD transporter shares a high homology with AcrB but its substrate profile is narrower [29]. AcrD does not possess its own periplasmatic adaptor, but it depends on AcrA to form a tripartite complex [30].

The infectious strategy of *Shigella* relies on its capacity to invade the colonic mucosa where it induces a strong inflammatory response, provoking a massive destruction of the epithelium [31]. *Shigella* crosses the intestinal barrier by M-cell transcytosis, then is exposed to and phagocytosed by resident macrophages, from which it rapidly escapes, inducing their inflammatory cell death. Once released from dying macrophages, *Shigella* invades colonocytes through the basolateral side and, following the lysis of the phagocytic vacuole, replicates freely in the cytosol. Then, thanks to actin-based motility, it spreads from one cell to another in the epithelium without further extracellular steps. The colonization of intestinal epithelia is carried out by the coordinated action of several virulence factors controlling both the invasion process and the host’s innate response. Most of the genes required for the invasion process, including a Type III secretion system and its effectors, are located on a large virulence plasmid and are coordinately expressed in response to the host environment signals [31,32,33].

Despite the relevance of AcrAB in several pathogens, the only data available for *Shigella* indicate that AcrAB is induced in the presence of bile salts and is required for biofilm formation [34].

In this work, we sought to understand whether AcrAB and/or AcrD, the two MDR EPs of the RND family conserved in *S. flexneri* M90T, impact on the *Shigella* pathogenicity process. We monitored how the loss of the AcrB or AcrD inner membrane transporters and of the AcrA periplasmatic component affects the capability of *S. flexneri* to invade and survive within macrophages and epithelial cells. The results we obtained clearly indicate that a functional AcrAB efflux pump is required during the invasion of epithelial cells.

## 2. Materials and Methods

### 2.1. Bacterial Strain and Plasmid Construction

Bacterial strains used in this study are listed in Table 1. *E. coli* DH10b has been used as the recipient in cloning experiments. To construct *acrAB*, *acrA*, and *acrB* deletion mutants of M90T, gene disruption was performed using the one-step inactivation method of chromosomal genes [35]. In particular, the kanamycin resistance gene was amplified via PCR using pKD4 as template and the oligo pairs ShAF/ShBR for the *acrAB* deletion, ShAF/ShAR for the *acrA* deletion and ShBF/ShBR for the *acrB* deletion (Appendix A). The resulting PCR products were used to transform M90T recipient strain harboring the pKD46 plasmid expressing the Red recombinase. In order to maintain unaltered transcription and translation of the downstream *acrB* gene, the Km resistance cassette in the mutant M90T Δ*acrA* was eliminated through the flippase encoded by the pCP20 plasmid using flippase/flippase recognition target (Flp/FRT) recombination [35]. To construct the M90T Δ*acrD* mutant, P1 phage isolated from the donor strain JW2454-1 *∆acrD* of the Keio Collection [36] was used to infect M90T wild-type strain and transduction of the mutated locus was verified using PCR. All plasmids used in this study are listed in Table 1. Plasmid pGSf*acrB* was obtained by cloning *acrB* coding sequence downstream the tac promoter of pGIP7 plasmid and then used as template for site directed mutagenesis. In particular, as previously described [28], the D408A substitution in AcrB encoded by the pSf*acrB*_D408A_ plasmid was obtained via a Site-Directed Mutagenesis System (GENEART^®^ Invitrogen-Thermo Fisher Scientific, Waltham, MA, USA), using the oligo pair pSf*acrB*_D408A_F/pSf*acrB*_D408A_R (Appendix A), that allowed to replace the A at position 1223 (from the ATG of *acrB*) with C, generating a codon change from GAC to GCC (D408A). The presence of the correct mutation on pSf*acrB*_D408A_ plasmid has been verified via DNA sequencing (Biofab, Rome, Italy). The obtained plasmid was then transformed in the M90T Δ*acrB* strain.

### 2.2. General Procedures and Growth Conditions

PCR reactions were routinely performed using DreamTaq DNA polymerase (Thermo Fisher Scientific, Waltham, MA, USA) for screening assays or Pfu Taq DNA polymerase (Thermo Fisher Scientific, Waltham, MA, USA) for amplification of coding sequences used for cloning purposes. Mutations and constructs obtained in this study were verified via DNA sequencing (BioFab, Rome, Italy).

Unless otherwise indicated, bacteria were grown aerobically in LB medium at 37 °C. Solid media contained 1.6% agar. Congo red was added (0.01% final concentration) to the Trypticase soy agar to monitor the expression of the virulence phenotype prior to infection assays. Antibiotics were used at the following concentrations: ampicillin 50 μg/mL; cloramphenicol 25 μg/mL; kanamycin 30 μg/mL; streptomycin 10 μg/mL. Growth kinetics of M90T and its derivatives were measured using a CLARIOstar plate reader (BMG LABTECH, Offenburg, Germany).

### 2.3. Antimicrobial Susceptibility

In order to determine MIC of erythromycin, tetracycline and streptomycin, M90T wild-type strain and its derivatives M90T Δ*acrAB*, M90T Δ*acrA*, M90T Δ*acrB* pSf*acrB*_D408A_, and M90T Δ*acrD*, were inoculated into LB and grown at 37 °C by shaking for 16 h. Cultures were then diluted to OD_600_ 0.02 in LB and 100 µL aliquots were transferred to 96-well plate, each well containing 100 µL of 2-fold serial dilutions of the compounds to be tested (erythromycin 0.001 mg/mL to 2 mg/mL, tetracycline 0.0078 μg/mL to 16 μg/mL, and streptomycin 0.0625 μg/mL to 128 μg/mL). After 16 h incubation at 37 °C, the lowest concentration of antibiotic inhibiting bacterial growth was estimated, as previously described [40]. At least three biological replicates were performed. Bile salts (B8756, Sigma-Aldrich, St. Louis, MO, USA), consisting of an approximate 1:1 mixture of cholate and deoxycholate, were used at 0.25% (*w*/*v*).

### 2.4. Cell Culture and Infection

Infection experiments were performed using THP-1 (TIB-202, American Type Culture Collection, Manassas, VA, USA) and Caco-2 (HTB-37, American Type Culture Collection, Manassas, VA, USA) cell lines. THP-1 cells were grown in RPMI 1640 (Gibco, Thermo Fisher Scientific, Waltham, MA, USA) medium containing 10% heat-inactivated fetal bovine serum (FBS) (Euroclone S.P.A., Milan, Italy), 2 mM L-glutamine and PS (0.05 I.U./mL penicillin and 0.05 I.U./mL streptomycin), referred to as RF10, at 37 °C in a humidified 5% CO_2_ atmosphere. Before bacterial infection, THP-1 monocytes were differentiated into macrophages as previously described [41]. Two hours before bacterial addition, THP-1 derived macrophages were fed fresh RPMI containing only L-glutamine. The human epithelial colorectal adenocarcinoma Caco-2 cell line was grown in Dulbecco minimal essential medium (DMEM) (Gibco, Thermo Fisher Scientific, Waltham, MA, USA) containing 10% FBS and PS, referred to as DF10, at 37 °C in a humidified 5% CO_2_ atmosphere. For bacterial infection, cells were seeded in 6-well tissue culture plates (Falcon, Thermo Fisher Scientific, Waltham, MA, USA) at 4.0 × 10^5^ cells/well in DF10. After 24 h, cells were serum-starved overnight in DMEM supplemented with 0.5% FBS and PS (DF0.5). Two hours before bacterial infection, DF0.5 was replaced with fresh DMEM containing only L-glutamine. Both cell lines were infected at MOI 100. The plates were then centrifuged (15 min, 750× *g*) and incubated for 30 min (THP-1) or 45 min (Caco-2) at 37 °C under 5% CO_2_ atmosphere to allow the bacteria to enter the cells. Finally, extracellular bacteria were removed by washing three times with 1× PBS. This point was taken as time zero (T0). Fresh medium (RPMI or DMEM) containing gentamicin (100 μg/mL) was added to kill extracellular bacteria, and infected cells were incubated at 37 °C up to 3 (THP-1 infection) or 4 h (Caco-2 cells infection). When indicated, M90T bacteria were treated with 100 μg/mL of 1-(1-naphthyl-methyl)-piperazine (NMP) (Sigma-Aldrich, St. Louis, MO, USA) just before adding to the host cells.

### 2.5. Live and Death Assay and Viable Bacterial Count

To collect intracellular bacteria at each infection time point, infected cells were washed twice with 1× PBS and lysed by adding 1% Triton X-100 (Sigma-Aldrich, St. Louis, MO, USA)) for 5 min. To evaluate the percentage of intracellular dead bacteria, recovered bacteria were pelleted, washed with 1× PBS and suspended in 1× PBS containing 10 μg/mL DAPI to stain the entire population and 15 μM Propidium Iodide (PI) to label dead bacteria. After 20 min of incubation at room temperature in the dark, bacteria were washed with 1× PBS and resuspended in 5 μL 1× PBS containing 50% glycerol. The entire sample was spotted on the glass slide and overlaid with the coverslip for immediate observation and counting under the fluorescence microscopy. Samples were examined using a Leica DMRE fluorescence microscope equipped with a 100× lens.

To define the number of viable intracellular bacteria, infection of epithelial cells was carried out as describe above. At the indicated time points, the cell lysate containing intracellular bacteria was collected, washed and resuspended in 1× PBS. Serial dilutions of the bacterial suspensions were plated on LB agar plates to calculate the CFU/mL.

### 2.6. Plaque Assay

To determine the ability of the M90T wild-type strain and its derivatives to spread intercellularly a plaque assay was performed [42]. The 5 × 10^6^ Caco-2 cells were seeded in 60 mm plates in DF10. Once reached confluency (usually after 24 h), cells were serum-starved overnight in DF0.5. DF0.5 medium was replaced with DMEM containing only L-glutamine 2 h before the infection. The infection was carried out at MOI 0.001, plates were centrifuged at 750 g for 15 min and subsequently incubated at 37 °C in a 5% CO_2_ atmosphere for 45 min. Extracellular bacteria were then removed by washing the plates three times with 1× PBS. An agarose overlay containing DMEM, gentamicin (100 μg/mL), FBS (5%) and agarose (0.5%) was added to each plate. Infected cells were incubated for 72 h, then the agarose overlay was carefully removed, and cells were ethanol fixed and Giemsa stained.

### 2.7. LDH Cytotoxicity Assay

CyQUANT™ LDH Cytotoxicity Assay kit (Invitrogen, Thermo Fisher Scientific, Waltham, MA, USA) was used to measure cytotoxicity and verify that the NMP inhibitor does not affect cell viability. Caco-2 cells were plated in DMEM with or without NMP (100 μg/mL) in 35 mm plates (Falcon, Thermo Fisher Scientific, Waltham, MA, USA) at 0.8 × 10^6^ cells/well. LDH activity was determined after 2 and 4 h of treatment by measuring absorbance at 490 nm and 680 nm with a CLARIOstar plate reader (BMG LABTECH, Offenburg, Germany). Percentage of cytotoxicity was calculated according to the manufacturer’s instructions.

### 2.8. Statistical Analysis

Statistically significant differences in viable bacterial counts and live and death assays were identified using a two-tailed student’s *t*-test.

## 3. Results

### 3.1. Lack of AcrAB Impairs the S. flexneri M90T Infection of Epithelial Cells

AcrAB is one of the most important MDR EP of *E. coli* and is expressed at high levels even under laboratory conditions [24]. Despite the intense gene decay of the housekeeping genes [20], the *acrA* and *acrB* encoding genes are conserved in all *Shigella* spp. Additionally, they share a high homology with those of *E. coli* [4]. As for *E. coli*, *acrAB* operon is also highly expressed in *S. flexneri* M90T. During the infection of U937-derived macrophages and Caco-2 epithelial cells the mRNA level of *acrA* gene, measured to monitor the behavior of the entire *acrAB* operon, further increases very early upon *Shigella* entry into host cells, the transcriptional response being more evident in M90T infecting Caco-2 cells [11]. Based on these notions and on the role this MDR EP plays in other pathogenic bacteria, we wondered whether AcrAB can be part of the mechanisms ensuring a successful *Shigella* infection process. To this end, an M90T derivative lacking the entire *acrAB* operon was generated via the one-step method of gene inactivation [35] (Table 1). Lack of AcrAB does not have an impact on M90T growth properties in laboratory conditions (Appendix A), while, as expected, it causes a higher susceptibility to drugs as compared to the M90T wild-type strain (Figure 1A). Silencing of *acrAB* also affects M90T resistance to bile salts at physiological concentration (0.25% *w*/*v*) (Figure 1B), which is in agreement with previous observations [34].

To test the response to the host cell environment in the absence of AcrAB, both THP-1- derived macrophages and Caco-2 epithelial cells were infected with the wild-type M90T strain and the Δ*acrAB* derivative. We first assessed *Shigella* intracellular survival via DAPI/PI double staining of bacteria harvested from infected cells at different time points post-infection (p.i.). Figure 2A shows that lack of AcrAB EP poorly affects the viability of M90T when inside macrophage environment as, by and large, the proportion of dead bacteria detected at each time point was very similar either for the wild-type or mutant strain. Conversely, the absence of AcrAB is detrimental to M90T infecting epithelial cells. Indeed, as shown in Figure 2B, while a certain fraction of dead wild-type bacteria was observed to constantly increase throughout the infection period analyzed, a high amount of PI-positive M90T Δ*acrAB* was found already at a very early stage of infection (T0) of Caco-2 cells. This proportion further increases at two hours p.i., keeping higher than that of the parental strain till the last time point analyzed.

This observation prompted us to explore in more details the behavior of Δ*acrAB* derivative during the infection of epithelial cells. We determined the viability of bacteria recovered from infected epithelial cells using the CFU assay. We found that, as expected, the number of colonies formed by intracellular mutant bacteria harvested at the various time points is much lower compared to M90T. However, the CFU/mL produced by the Δ*acrAB* derivative increases over the infection period following a kinetics very similar to that of the parental strain (Figure 3A) suggesting that, although much more Δ*acrAB* cells succumb to the host attacks, the surviving intracellular bacteria do not lose the ability to multiply. Full invasion process relies on *Shigella* capability to disseminate to neighboring cells in an epithelial layer without further extracellular step [31]. The cell-to-cell spreading capacity of viable Δ*acrAB* was investigated using plaque assay. Figure 3B shows that lack of AcrAB dramatically impairs the spreading of M90T to adjacent Caco-2 cells as a very low number of plaques formed by the *acrAB* deletion mutant was observed, representing less than 3% of the parental strain.

### 3.2. Both AcrB and AcrA Components Contribute to the Survival of S. Flexneri M90T Inside Epithelial Cells

It is well acknowledged that each of the components of the AcrAB-TolC MDR EP has specific activities ensuring the proper exporter function, as the inner membrane transport protein AcrB is critical for the substrate specificity and the periplasmic adapter protein AcrA is important for connecting AcrB to the TolC channel [12]. Based on the results obtained with the M90T Δ*acrAB* strain we asked whether both AcrB and AcrA are required for the survival of *Shigella* inside epithelial cells. To this end, single *acrA* and *acrB* deletion mutants were generated via site-directed mutagenesis (Table 1). Since AcrB is a very abundant inner membrane protein, the M90T Δ*acrB* strain was complemented with an AcrB protein lacking the efflux activity (AcrB D408A) (Table 1), in order to generate a defective strain for AcrB-associated transporter function without altering the membrane protein composition. Moreover, based on the knowledge that the AcrA periplasmic adapter is also used by AcrD [30], the other RND EP conserved in M90T, to form a functional transporter complex, a Δ*acrD* strain has been also constructed (Table 1). M90T Δ*acrA,* M90T Δ*acrB* pSf*acrB*_D408A_ and M90T Δ*acrD* displayed growth properties overlapping those of M90T wild-type strain (Appendix A). Both M90T Δ*acrA* andM90T Δ*acrB* pSf*acrB*_D408A_ were much more susceptible than M90T to the antibiotics tested, while lack of AcrD did not change the MIC profile of M90T (Figure 1A). The presence of bile salts in the LB medium affected the growth curve of all the three M90T derivatives, with milder effects on the multiplication rate of M90T Δ*acrB* pSf*acrB*_D408A_ and M90T Δ*acrD* (Figure 1B). Parallel Caco-2 epithelial cell infections were carried out with the three mutant strains along with wild-type M90T and the viability of intracellular bacteria was assessed at different time points p.i. using DAPI/PI double staining. As shown in Figure 4A, the percentage of M90T Δ*acrB* pSf*acrB*_D408A_ PI-positive bacteria was very similar to the wild-type at very early stage of the infection (T0), while it significantly increased in the following hours of infection. The defective phenotype conferred by the absence of AcrA was much more severe, chiefly soon after bacterial entry into the host cells (T0). Indeed, at this time point, we found that more than 20% of intracellular M90T Δ*acrA* bacteria were not viable compared to 2% observed for the intracellular wild-type strain. The proportion of dead M90T Δ*acrA* bacteria further increases as the infection proceeds. Data obtained for the M90T strain lacking AcrD indicate that this transporter is dispensable for *Shigella* survival inside the epithelial cells. Indeed, the proportion of PI-positive M90T Δ*acrD* bacteria was very similar to that of the parental strain at each time point analyzed (Figure 4A). Overall, the data obtained using deletion mutants in single components of the two RND EPs point out the main role of the AcrAB EP during the *Shigella* infection of the epithelial cells.

AcrAB inhibitors exist, mainly targeting the transporter function of the AcrB protein [1]. Considering the results we obtained, it is reasonable to regard such compounds as potential anti-*Shigella* virulence. Indeed, by inhibiting AcrB-mediated extrusion, survival, and thus dissemination of the pathogen should be significantly impaired. To test this hypothesis, we carried out infection experiments in the presence of 1-(1-naphthyl-methyl)-piperazine (NMP), a well characterized and active AcrB inhibitor [1,43]. This inhibitor belongs to the family of arylpiperazines and it has been suggested to block the normal substrate extrusion by acting as an EP substrate [1]. NMP neither affects M90T viability/growth properties nor has toxic effects on Caco-2 cells, the latter assessed by measuring LDH activity (Lactate dehydrogenase) present in the cell supernatant (data not shown). Caco-2 cells were infected with the M90T wild-type strain in the presence or absence of NMP. The drug (100 µg/mL) was added to the bacteria in DMEM medium, just before exposing them to the epithelial cells and the effect of the treatment on viability was monitored soon after bacterial entry into the host cells (T0) and after two hours of infection. As shown in Figure 4B, drug treatment mildly affects the survival of intracellular bacteria at a very early stage of infection, while it causes a significant decrease in intracellular M90T viability at two hours p.i. Interestingly, the M90T phenotype induced by the AcrB inhibitor overlaps with that observed with the intracellular Δ*acrB* p*acrB*_D408A_ derivative at the same time points, highlighting, on one hand, the inhibitor specificity and confirming, on the other hand, the important involvement of AcrB transporter in assisting *Shigella* during the infection of epithelial cells.

## 4. Discussion

In this work, we report evidence demonstrating that the MDR EP AcrAB contributes to the virulence of *S. flexneri* by favoring the bacterial survival within the epithelial cells. Analysis of mutants lacking AcrA or AcrB clearly indicates that both components are required for the bacterial invasion of and survival within the Caco-2 epithelial cells. Moreover, the loss of AcrAB hampers the ability of *S. flexneri* to spread within the epithelial monolayer.

The hallmark of *Shigella* pathogenicity is its ability to penetrate the colonic epithelium, escape macrophages by inducing cell death, and, subsequently, invade colonocytes from the basolateral side and propagate infection through cell-to-cell spread, eventually causing the destruction of the intestinal barrier function [31]. The whole infection process is mostly built on the action of several factors encoded by the virulence plasmid that *Shigella* acquired during the evolution [18,19]. On the road of evolution from commensal *E. coli* ancestor, *Shigella* has also undergone a patho-adaptation process following an intense gene decay involving the loss of detrimental or unnecessary functions for intracellular lifestyle [18,20,21,22,23]. Gene silencing also affected some MDR EPs belonging to the RND family, as, of those present in *E. coli* K12 MG1655, only the *acrAB* and *acrD* genes were conserved as fully functional in the *S. flexneri* M90T genome [11]. Among these two, AcrAB undoubtedly stands out for its important contribution to the pathogenicity of several bacteria [27,44,45,46,47]. More recently, the pivotal role of AcrAB in supporting survival of AIEC LF82 within the macrophages has been also demonstrated [28], further expanding the range of bacterial pathogens exploiting AcrAB function to facilitate the infection process. Regarding *Shigella*, the involvement of AcrAB in the invasion of host cells has never been explored. The data obtained from the present study clearly indicate that the AcrAB function protects *Shigella* during epithelial cell infection, mainly at the early stages. Indeed, lack of the entire *acrAB* operon results in a larger proportion of intracellular bacteria undergoing cell death, compared to the wild-type strain, soon after cell entry, picking at two hours p.i. We also show that surviving intracellular M90T Δ*acrAB* bacteria appear perfectly viable and multiply with a kinetics similar to that of the parental M90T. However, when the Δ*acrAB* derivative was challenged for the ability to disseminate within the Caco-2 monolayer, most of the viable bacteria failed to form visible plaques, at least in the time taken by the wild-type M90T. Indeed, as can be inferred by comparing the CFU/mL with the number of plaques (Figure 3), the viable intracellular M90T Δ*acrAB* bacteria forming colonies on LB agar are about one-seventh of the parental strain, while those forming plaques are less than the thirtieth part of the wild-type M90T. This observation suggests that the absence of AcrAB, in addition to causing accumulation of deadly toxic metabolites, might also interfere with the expression of functions involved in the actin-based intra/intercellular movement of *Shigella*. In this regard, it is interesting to consider transcriptomic studies carried out in *Salmonella* demonstrating that, in the *acrAB*-defective background several genes essential for invasion, encoded by the *Salmonella* pathogenicity island (SPI), are downregulated [27,48]. Moreover, it has been recently suggested that the enrichment of certain natural AcrB substrates in the exometabolome might be important for *Salmonella* virulence, some of these being involved in the regulation of virulence factor expression [49]. Assessing whether modulation of virulence factors also occurs in *Shigella* AcrAB-defective mutant will deserve a much deeper investigation in the future.

It is well known that AcrA and AcrB provide the EP with specific functions contributing to its full activity. The evidence presented here indicates that the inactivation of AcrA alone leads to a marked decrease in bacterial viability in epithelial cells comparable to that observed with the M90T double mutant. As discussed above, among the MDR EPs belonging to the RND family, *S. flexneri* M90T has also kept functional AcrD transporter, which is orphaned of its own periplasmic protein and shares AcrA with the AcrB transporter [11,30]. The evaluation of the impact of each transporter indicates that the activity associated with AcrB is strongly required by *Shigella* to better survive within epithelial cells, while AcrD appears to be dispensable, at least for the intracellular viability. The use of a M90T Δ*acrB* derivative complemented with the unfunctional AcrB D408A protein in the infection experiments of epithelial cells allowed us to avoid profound alteration of the inner membrane due to the lack of an abundant constituent, such as AcrB and to specifically pinpoint the involvement of the efflux activity associated with AcrB.

To date, this is the second study emphasizing the role that EPs play in supporting *Shigella* to overcome the hostile environment met inside host cells. Depending on the host cell context the activity of different EPs is exploited. It has been recently demonstrated that the MFS EmrKY is specifically activated in macrophages in response to the cytoplasmic acidic pH induced by *Shigella* infection, improving the bacterial fitness [11]. Here, we show that the highly expressed AcrAB works to promote invasion of and spread within epithelial cells and that both AcrA and AcrB components are required. Overall, these reports point out EPs as important components of the *Shigella* pathogenicity mechanism, and, together with the data we obtained with the AcrB specific inhibitor, lead to envisage the efflux activity of specific transporters and/or the expression of specific EPs as valuable targets to attenuate *Shigella* virulence.

## Figures and Tables

**Figure 1 biomolecules-13-00823-f001:**
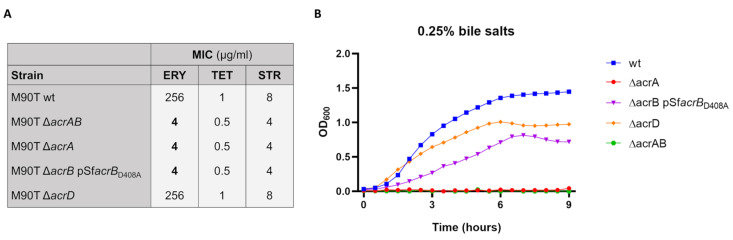
Susceptibility of M90T strains to antibiotics and bile salts. (**A**) MICs of erythromycin (ERY), tetracycline (TET) and streptomycin (STR) for M90T wild-type and mutant strains. Values in bold indicate a significant decrease (>2-fold) compared with those of the parental strain. (**B**) Effect of bile salts on the growth of *S. flexneri* M90T and its derivatives. Bacterial growth was monitored in LB medium containing 0.25% (*w*/*v*) of bile salts up to 9 h. The growth curve shown derives from one of three experiments, which gave similar results.

**Figure 2 biomolecules-13-00823-f002:**
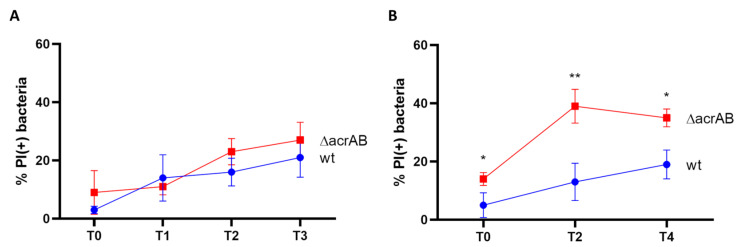
AcrAB efflux pump differentially contributes to the survival of *S. flexneri* M90T within (**A**) THP-1-derived macrophages, where no significative difference can be observed at 0, 1, 2 and 3 h p.i. (referred to as T0, T1, T2 and T3, respectively), and (**B**) Caco-2 epithelial cells, where the lack of *acrAB* significantly affects the viability of intracellular M90T at 0, 2 and 4 h p.i. (referred to as T0, T2 and T4, respectively). The percentages of intracellular M90T wild-type (blue curve) and Δ*acrAB* (red curve) PI (+) dead bacteria relative to DAPI (+) bacteria are shown. The results are the average of at least three independent experiments. Error bars represent the SD. The statistical significance was determined using a two tailed student’s *t*-test. * *p* ≤ 0.01; ** *p* ≤ 0.001.

**Figure 3 biomolecules-13-00823-f003:**
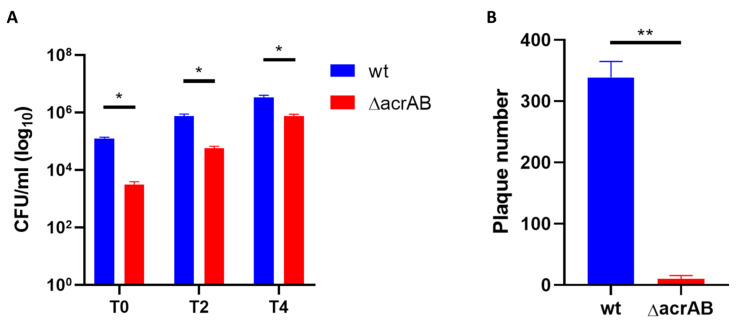
Lack of AcrAB affects the spreading ability of viable M90T within an epithelial cell monolayer. (**A**) Number of viable intracellular bacteria was determined using CFU assay after Caco-2 cells infection with M90T wild-type and Δ*acrAB* strains. Infected Caco-2 cells were lysed at the indicated time points (T0, T2 and T4) and serial dilutions of intracellular bacteria plated on LB agar. (**B**) Confluent Caco-2 cells were infected with M90T wt and Δ*acrAB* at MOI 0.001. Cells were stained with Giemsa and number of plaques was counted 72 h post-infection. The results shown are the average of at least three independent experiments. Error bars represent the SD. The statistical significance was determined using a two tailed student’s *t*-test. * *p* ≤ 0.01; ** *p* ≤ 0.001.

**Figure 4 biomolecules-13-00823-f004:**
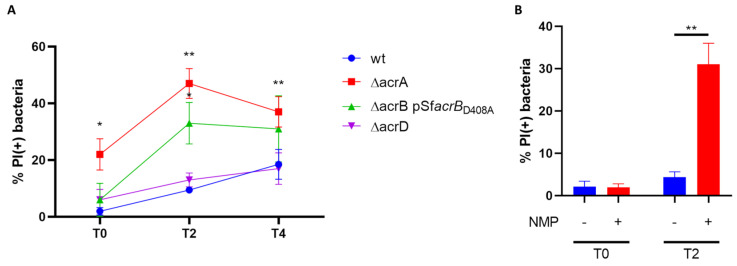
Both AcrA and AcrB component contribute to the M90T viability within epithelial cells. (**A**) Live and dead assay was performed on intracellular M90T wt (blue curve), M90T Δ*acrA* (red curve), M90T Δ*acrB* complemented with the unfunctional AcrB D408A transporter (pSf*acrB*_D408A_) (green curve) and M90T Δ*acrD* (purple curve) bacteria recovered from Caco-2 cells at 0, 2 and 4h p.i. (referred to as T0, T2, and T4, respectively). The values are expressed as percentage of PI (+) dead bacteria relative to DAPI (+) bacteria. (**B**) Inhibition of AcrB function by treatment with NMP affects *S. flexneri* survival within epithelial cells. Caco-2 cells were infected with M90T wt untreated (blue bar) or treated with 100 μg/mL NMP (red bar) and intracellular bacteria were recovered at the beginning (T0) and after 2h of infection (T2). The results are the average of at least three independent experiments. Error bars represent the SD. The statistical significance was determined using a two tailed student’s *t*-test. * *p* ≤ 0.01; ** *p* ≤ 0.001.

**Table 1 biomolecules-13-00823-t001:** Strains and plasmids used in this study.

Strain	Relevant Characteristics	Source/Reference
DH10b	F– *mcrA* Δ(*mrr*-*hsdRMS*-*mcrBC*) φ80*lacZ*ΔM15 Δ*lacX*74 *recA*1 *endA*1 *araD*139 Δ (*ara*-*leu*)7697 *galU galK* λ– *rpsL*(StrR) *nupG*	[37]
JW2454-1 ∆*acrD*	F-, Δ(*araD*-*araB*)567, ΔlacZ4787(::rrnB-3), λ-, Δ*acrD*790::kan, rph-1, Δ(*rhaD*-*rhaB*)568, hsdR514. Km ^R^ (Resistant)	[36]
M90T	M90T *S. flexneri* 5a	[38]
M90T Δ*acrAB*	M90T derivative defective in *acrAB* operon, Km ^R^	This study
M90T Δ*acrA*	M90T derivative defective in *acrA* gene, Km ^S^ (Sensitive)	This study
M90T Δ*acrB*	M90T derivative defective in *acrB* gene, Km ^R^	This study
M90T Δ*acrD*	M90T derivative defective in *acrD* gene, Km ^R^	This study
**Plasmid**	**Relevant Characteristics**	**Source/Reference**
pKD46	Red recombinase expression plasmid, Ap ^R^	[35]
pKD4	Template plasmid carrying a kanamycin resistance gene flanked by Flp recognition target sequences, Km ^R^, Ap ^R^	[35]
pCP20	Temperature sensitive replicon carrying the yeast Flp recombinase gene, Ap ^R^	[35]
pGIP7	pACYC184-derived vector carrying lacI-lac promoter region, Cm ^R^	[39]
pGSf*acrB*	pGIP7 derivative plasmid carrying the *acrB* gene in *BamHI* site, Cm ^R^	This study
pSf*acrB*_D408A_	pGIP7 derivative plasmid carrying the *acrB*_D408A_ allele, Cm ^R^	This study

## Data Availability

The data sets used and/or analyzed during the current study are available from the corresponding author upon reasonable request.

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
