# Peer review of "Role of the MDR Efflux Pump AcrAB in Epithelial Cell Invasion by Shigella flexneri"

_biomolecules, 2023, doi:10.3390/biom13050823_

Round 1
Reviewer 1 Report
In my view of the article submitted by Dr. Martina Pasqua et al. and her colleagues is an elegant, well-designed and well-written paper, which highlights the role of AcrAB-TolC efflux pump and its ability to extrude agents and compounds harmful to bacteria has not only in underlying antimicrobial resistance, but rather playing a central role in the virulence. Maybe this is the reason why a highly specialized E. coli has maintained such genes, I think the article deserves to be published.
on line 45 “the E.coli specie” should be species
on line 280 epithelia should be epithelial
I have enjoyed reading this paper
Author Response
We thank the reviewer for the very positive evaluation of our work.
On line 45 “the E.coli specie” should be species
Done.
On line 280 epithelia should be epithelial
Done.
Reviewer 2 Report
The role of the Shigella flexneri AcrAB resistant efflux pump in epithelial cell invasion was reported in this manuscript, providing further knowledge of AcrAB EP in Shigella flexneri. I support the publication of this article in „Biomolecules” after a minor revision and responding to the following reviewer´s comments:
1. Why did you use ampicillin, chloramphenicol, kanamycin, and streptomycin in your study? Why were the following concentrations chosen („Antibiotics were used at the following concentrations: ampicillin 50 μg/ml; chloramphenicol 25 μg/ml; kanamycin 30 μg/ml; streptomycin 10 μg/ml”)?
2. Please give more details about the NMP (1-(1-naphthyl-methyl)-piperazine) inhibitor!
3. Check typos and grammatical errors throughout the manuscript.
Author Response
We thank the reviewer for supporting our paper.
Why did you use ampicillin, chloramphenicol, kanamycin, and streptomycin in your study? Why were the following concentrations chosen („Antibiotics were used at the following concentrations: ampicillin 50 μg/ml; chloramphenicol 25 μg/ml; kanamycin 30 μg/ml; streptomycin 10 μg/ml”)?
Thank you for this observation. In agreement with the referee request we have modified Table 1 (Strains and Plasmids) adding the genotype of DH10b (SmR) and antibiotic resistance characteristic of the different plasmids used in this study (pKD46 ApR; pKD4 KmR; pCP20 ApR). The antibiotic concentrations used in this study are those routinely used in our laboratory and proved suitable for plasmid and strain selection.
Please give more details about the NMP (1-(1-naphthyl-methyl)-piperazine) inhibitor!
According to referee suggestion we have added a sentence in the Result section with more details on the NMP inhibitor (lane 317-319).
Check typos and grammatical errors throughout the manuscript.
We have carefully read the text and corrected typos and grammatical errors.
Reviewer 3 Report
In this study, Coluccia et al. study the role of RND family efflux pump AcrB in Shigella flexneri and it role in cell invasion. The AcrAB-TolC pump in Escherichia coli and other bacteria plays a major role in antibiotic resistance and also contributes to the virulence and pathogenesis of bacterial infections. This study shows that AcrAB specifically helps Shigella flexneri invade epithelial cells, and deletion of both acrA and acrB genes reduces the bacteria's survival within cells and prevents its spread. In general, I think this is a solid study with some interesting results, that I think would be a great addition to biomolecules.
I do have a couple of suggestions and a couple of questions for the authors to address.
1. Could the authors explain why the bile salts were only measured for 9 hours in Figure 1? It may be beneficial to obtain the minimum inhibitory concentration (MIC) for the bile salts used in this study. Additionally, could authors clarify which type of bile salts (cholate, taurocholate, etc.) were utilized?
2. Could the authors provide further insight into the molecular mechanism behind how the AcrAB-TolC system protects Shigella during epithelial cell infection?
Author Response
We thank the referee for appreciating our study.
Could the authors explain why the bile salts were only measured for 9 hours in Figure 1? It may be beneficial to obtain the minimum inhibitory concentration (MIC) for the bile salts used in this study. Additionally, could authors clarify which type of bile salts (cholate, taurocholate, etc.) were utilized?
Previous study (Nickerson 2017) has already observed the physiological range of bile salts in which Shigella is able to grow giving rise to a typical growth pattern. Also the increased sensitivity of S. flexneri acrB mutant to bile salts has been reported in the same work (Nickerson et al., 2017). For this reason, we only briefly address this issue. In particular, we repeated the growth curve of Shigella flexneri M90T (wt and its derivatives) in conditions comparable to those previously used to analyze another S. flexneri strain (2457) acrB defective to ascertain that our mutants exhibit similar behaviour. Nickerson and coworkers have performed their experiments using a commercial mixture of bile salts (Sigma B8756 0.4% w/v) consisting of an approximate 1:1 mixture of cholate and deoxycholate and have followed the growth curve for 6 hours. We have performed the experiments using a comparable physiological concentration of bile salts (0.25% w/v) and the same mixture (Sigma) and we have monitored the growth for 9h to cover also the stationary phase. According to the referee request we have added information in Material and Methods about the bile salts used (lane 142-143) and specified the physiological concentration used in the text.
Could the authors provide further insight into the molecular mechanism behind how the AcrAB-TolC system protects Shigella during epithelial cell infection?
The referee raises a very important point. In the Discussion section (from 379 to 385) we have already discussed the potential involvement of AcrAB in Shigella virulence. However, to better highlight the importance of this efflux pump we have added a brief discussion of recent data obtained by metabolomic analysis of natural AcrB substrates in Salmonella and a new reference (Wang-Kan et al., 2021) (lane 385-388).